# Learning to Execute Programs with Instruction Pointer Attention Graph Neural Networks

**David Bieber**
Google
dbieber@google.com

**Charles Sutton**
Google
charlessutton@google.com

**Hugo Larochelle**
Google
hugolarochelle@google.com

**Daniel Tarlow**
Google
dtarlow@google.com

## Abstract

Graph neural networks (GNNs) have emerged as a powerful tool for learning software engineering tasks including code completion, bug finding, and program repair. They benefit from leveraging program structure like control flow graphs, but they are not well-suited to tasks like program execution that require far more sequential reasoning steps than number of GNN propagation steps. Recurrent neural networks (RNNs), on the other hand, are well-suited to long sequential chains of reasoning, but they do not naturally incorporate program structure and generally perform worse on the above tasks. Our aim is to achieve the best of both worlds, and we do so by introducing a novel GNN architecture, the Instruction Pointer Attention Graph Neural Network (IPA-GNN), which achieves improved systematic generalization on the task of learning to execute programs using control flow graphs. The model arises by considering RNNs operating on program traces with branch decisions as latent variables. The IPA-GNN can be seen either as a continuous relaxation of the RNN model or as a GNN variant more tailored to execution. To test the models, we propose evaluating systematic generalization on learning to execute using control flow graphs, which tests sequential reasoning and use of program structure. More practically, we evaluate these models on the task of learning to execute partial programs, as might arise if using the model as a heuristic function in program synthesis. Results show that the IPA-GNN outperforms a variety of RNN and GNN baselines on both tasks.

## 1   Introduction

Static analysis methods underpin thousands of programming tools from compilers and debuggers to IDE extensions, offering productivity boosts to software engineers at every stage of development. Recently machine learning has broadened the capabilities of static analysis, offering progress on challenging problems like code completion [5], bug finding [2, 13], and program repair [25]. Graph neural networks in particular have emerged as a powerful tool for these tasks due to their suitability for learning from program structures such as parse trees, control flow graphs, and data flow graphs.

These successes motivate further study of neural network models for static analysis tasks. However, existing techniques are not well suited for tasks that involve reasoning about program execution. Recurrent neural networks are well suited for sequential reasoning, but provide no mechanism for learning about complex program structures. Graph neural networks generally leverage local program structure to complete static analysis tasks. For tasks requiring reasoning about program execution, we expect the best models will come from a study of both RNN and GNN architectures. We design

a novel machine learning architecture, the Instruction Pointer Attention Graph Neural Network (IPA-GNN), to share a causal structure with an interpreter, and find it exhibits close relationships with both RNN and GNN models.

To evaluate this model, we select two tasks that require reasoning about program execution: full and partial program execution. These "learning to execute" tasks are a natural choice for this evaluation, and capture the challenges of reasoning about program execution in a static analysis setting. Full program execution is a canonical task used for measuring the expressiveness and learnability of RNNs [29], and the partial program execution task aligns with the requirements of a heuristic function for programming by example. Both tasks require the model produce the output of a program, without actually running the program. In full program execution the model has access to the full program, whereas in partial program execution some of the program has been masked out. These tasks directly measure a model's capacity for reasoning about program execution.

We evaluate our models for systematic generalization to out-of-distribution programs, on both the full and partial program execution tasks. In the program understanding domain, systematic generalization is particularly important as people write programs to do things that have not been done before. Evaluating systematic generalization provides a strict test that models are not only learning to produce the results for in-distribution programs, but also that they are getting the correct result because they have learned something meaningful about the language semantics. Models that exhibit systematic generalization are additionally more likely to perform well in a real-world setting.

We evaluate against a variety of RNN and GNN baselines, and find the IPA-GNN outperforms baseline models on both tasks. Observing its attention mechanism, we find it has learned to produce discrete branch decisions much of the time, and in fact has learned to execute by taking short-cuts, using fewer steps to execute programs than used by the ground truth trace.

We summarize our contributions as follows:

- We introduce the task of learning to execute assuming access only to information available for static analysis (Section 3).

- We show how an RNN trace model with latent branch decisions is a special case of a GNN (Section 4).

- We introduce the novel IPA-GNN model, guided by the principle of matching the causal structure of a classical interpreter.

- We show this outperforms other RNN and GNN baselines. (Section 5).

- We illustrate how these models are well-suited to learning non-standard notions of execution, like executing with limited computational budget or learning to execute programs with holes.

## 2 Background and Related Work

Static analysis is the process of analyzing a program without executing it [1]. One common static analysis method is control flow analysis, which produces a control flow graph [3]. Control flow analysis operates on the parse tree of a program, which can be computed for any syntactically correct source file. The resulting control flow graph contains information about all possible paths of execution through a program. We use a statement-level control flow graph, where nodes represent individual statements in the source program. Directed edges between nodes represent possible sequences of execution of statements. An example program and its control flow graph are shown in Figure 1.

When executing a program, an interpreter's *instruction pointer* indicates the next instruction to execute. At each step, the instruction pointer corresponds to a single node in the statement-level control flow graph. After the execution of each instruction, the instruction pointer advances to the next statement. When the control flow graph indicates two possible next statements, we call this a branch decision, and the value of the condition of the current statement determines which statement the instruction pointer will advance to.

Recurrent neural networks have long been recognized as well-suited for sequence processing [24]. Another model family commonly employed in machine learning for static analysis is graph neural networks [20], which are particularly well suited for learning from program structures [2]. GNNs have been applied a variety of program representations and static analysis tasks [2, 8, 13, 21, 23, 25, 28].

| $n$ | Source | Tokenization ($x_n$) | | | | Control flow graph ($n \rightarrow n'$) | $N_{\text{in}}(n)$ | $N_{\text{out}}(n)$ |
|---|---|---|---|---|---|---|---|---|
| 0 | `v0 = 23` | 0 | = | v0 | 23 | | $\emptyset$ | $\{1\}$ |
| 1 | `v1 = 6` | 0 | = | v1 | 6 | | $\{0\}$ | $\{2\}$ |
| 2 | `while v1 > 0:` | 0 | while > | v1 | 0 | | $\{1, 7\}$ | $\{3, 8\}$ |
| 3 | `v1 -= 1` | 1 | -= | v1 | 1 | | $\{2\}$ | $\{4\}$ |
| 4 | `if v0 % 10 <= 3:` | 1 | if <= % | v0 | 3 | | $\{3\}$ | $\{5\}$ |
| 5 | `v0 += 4` | 2 | += | v0 | 4 | | $\{4\}$ | $\{6\}$ |
| 6 | `v0 *= 6` | 2 | *= | v0 | 6 | | $\{5\}$ | $\{7\}$ |
| 7 | `v0 -= 1` | 1 | -= | v0 | 1 | | $\{4, 6\}$ | $\{2\}$ |
| 8 | `<exit>` | – | – | – | – | | $\{2, 8\}$ | $\{8\}$ |

Figure 1: **Program representation.** Each line of a program is represented by a 4-tuple tokenization containing that line's (indentation level, operation, variable, operand), and is associated with a node in the program's statement-level control flow graph.

Like our approach, the graph attention network (GAT) architecture [27] is a message passing GNN using attention across edges. It is extended by R-GAT [6] to support distinct edge types.

The task of learning to execute was introduced by [29], who applied RNNs. We are aware of very little work on learning to execute that applies architectures that are specific to algorithms. In contrast, *neural algorithm induction* [7, 9–12, 14, 15, 19] introduces architectures that are specially suited to learning algorithms from large numbers of input-output examples. The modeling principles of this work inspire our approach. The learning to execute problem is different because it includes source code as input, and the goal is to learn the semantics of the programming language.

Systematic generalization measures a model's ability to generalize systematically to out-of-distribution data, and has been an area of recent interest [4]. Systematic generalization and model design go hand-in-hand. Motivated by this insight, our architectures for learning to execute are based on the structure of an interpreter, with the aim of improving systematic generalization.

## 3 Task

Motivated by the real-world setting of static analysis, we introduce two variants of the "learning to execute" task: full and partial program execution.

### 3.1 Learning to Execute as Static Analysis

In the setting of machine learning for static analysis, models may access the textual source of a program, and may additionally access the parse tree of the program and any common static analysis results, such as a program's control flow graph. However, models may not access a compiler or interpreter for the source language. Similarly, models may not access dependencies, a test suite, or other artifacts not readily available for static analysis of a single file. Prior work applying machine learning for program analysis also operates under these restrictions [2, 13, 21, 22, 28, 29]. We introduce two static analysis tasks both requiring reasoning about a program's execution statically: full and partial program execution.

In full program execution, the model receives a full program as input and must determine some semantic property of the program, such as the program's output. Full program execution is a canonical task that requires many sequential steps of reasoning. The challenge of full program execution in the static analysis setting is to determine the target without actually running the program. In previous work, full program execution has been used as a litmus test for recurrent neural networks, to evaluate their expressiveness and learnability [29]. It was found that long short-term memory (LSTM) RNNs can execute some simple programs with limited control flow and $\mathcal{O}(N)$ computational complexity. In a similar manner, we are employing this variant of learning to execute as an analogous test of the capabilities of graph neural networks.

The partial program execution task is similar to the full program execution task, except part of the program has been masked. It closely aligns with the requirements of designing a heuristic function for programming by example. In some programming by example methods, a heuristic function informs the search for satisfying programs by assigning a value to each intermediate partial program as to

whether it will be useful in the search [16]. A model performing well on partial program execution can be used to construct such a heuristic function.

We consider the setting of bounded execution, restricting the model to use fewer steps than are required by the ground truth trace. This forces the model to learn short-cuts in order to predict the result in the allotted steps. We select the number of steps allowed such that each loop body may be executed at least twice. In our experiments, we use bounded execution by default. However, we compare in Section 5 to a Trace RNN model that follows the ground truth control flow, and surprisingly we find that the bounded execution IPA-GNN achieves better performance on certain length programs.

In both task settings we train the models on programs with limited complexity and test on more complex programs to evaluate the models for systematic generalization. This provides a quantitative indication of whether the models are not only getting the problem right, but getting it right for the right reasons – because they've learned something meaningful about the language semantics. We expect models that exhibit systematic generalization will be better suited for making predictions in real world codebases, particularly when new code may be added at any time. Another perhaps less appreciated reason to focus on systematic generalization in learning to execute tasks is that the execution traces of real-world programs are very long, on the order of thousands or even millions of steps. Training GNNs on such programs is challenging from an engineering perspective (memory use, time) and a training dynamics perspective (e.g., vanishing gradients). These problems are significantly mitigated if we can strongly generalize from small to large examples (e.g., in our experiments we evaluate GNNs that use well over 100 propagation steps even though we only ever trained with 10s of steps, usually 16 or fewer).

### 3.2 Formal Specification and Evaluation Metrics

We describe both tasks with a single formalization. We are given a complexity function $c(x)$ and dataset $D$ consisting of pairs $(x, y)$; $x$ denotes a program, and $y$ denotes some semantic property of the program, such as the program's output. The dataset is drawn from an underlying distribution of programs D, and is partitioned according to the complexity of the programs. $D_{\text{train}}$ consists only of examples $(x, y)$ satisfying $c(x) \leq C$, the threshold complexity, while $D_{\text{test}}$ consists of those examples satisfying $c(x) > C$. For each program $x$, both the textual representation and the control flow graph are known. $x_n$ denotes a statement comprising the program $x$, with $x_0$ denoting the start statement. $N_{\text{in}}(n)$ denotes the set of statements that can immediately precede $x_n$ according to the control flow graph, while $N_{\text{out}}(n)$ denotes the set of statements that can immediately follow $x_n$. Since branch decisions are binary, $|N_{\text{out}}(n)| \leq 2 \,\forall\, n$, while $|N_{\text{in}}(n)|$ may be larger. We define $N_{\text{all}}(n)$ as the full set of neighboring statements to $x_n$: $N_{\text{all}}(n) = N_{\text{in}}(n) \cup N_{\text{out}}(n)$. The rest of the task setup follows the standard supervised learning formulation; in this paper we consider only instances of this task with categorical targets, though the formulation is more general. We measure performance according to the empirical accuracy on the test set of programs.

## 4 Approach

In this section we consider models that share a causal structure with a classical interpreter. This leads us to the design of the Instruction Pointer Attention Graph Neural Network (IPA-GNN) model, which we find takes the form of a message passing graph neural network. We hypothesize that by designing this architecture to share a causal structure with a classical interpreter, it will improve at systematic generalization over baseline models.

### 4.1 Instruction Pointer RNN Models

When a classical interpreter executes a straight-line program $x$, it exhibits a simple causal structure. At each step of interpretation, the interpreter maintains a state consisting of the values of all variables in the program, and an instruction pointer indicating the next statement to execute. When a statement is executed, the internal state of the interpreter is updated accordingly, with the instruction pointer advancing to the next statement in sequence.

A natural neural architecture for modeling the execution of straight-line code is a recurrent neural network because it shares this same causal structure. At step $t$ of interpretation, the model processes

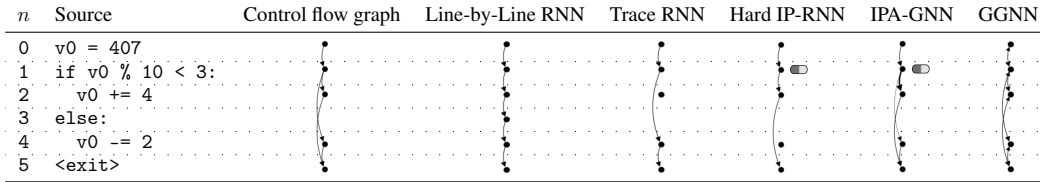

| $n$ | Source | Control flow graph | Line-by-Line RNN | Trace RNN | Hard IP-RNN | IPA-GNN | GGNN |
|---|---|---|---|---|---|---|---|
| 0 | `v0 = 407` | | | | | | |
| 1 | `if v0 % 10 < 3:` | | | | | | |
| 2 | `    v0 += 4` | | | | | | |
| 3 | `else:` | | | | | | |
| 4 | `    v0 -= 2` | | | | | | |
| 5 | `<exit>` | | | | | | |

Figure 2: **Model paths comparison.** The edges in these graphs represent a possible set of paths along which information may propagate in each model. The pills indicate the positions where a model makes a learned branch decision. The Hard IP-RNN's branch decision is discrete (and in this example, incorrect), whereas in the IPA-GNN the branch decision is continuous.

statement $x_{t-1}$ to determine its hidden state $h_t \in \mathbb{R}^H$, which is analogous to the values of the variables maintained in an interpreter's state. In this model, which we call the Line-by-Line RNN model, the hidden state is updated as

$$h_t = \text{RNN}(h_{t-1}, \text{Embed}(x_{n_{t-1}})), \tag{1}$$

where $n_t = t$ is the model's instruction pointer.

Each of the models we introduce in this section has the form of (1), but with different definitions for the instruction pointer $n_t$. We refer to these models as Instruction Pointer RNNs (IP-RNNs).

Consider next when program $x$ includes control flow statements. When a classical interpreter executes a control flow statement (e.g. an if-statement or while-loop), it makes a branch decision by evaluating the condition. The branch decision determines the edge in the program's control flow graph to follow for the new value of the instruction pointer. At each step of execution, values of variables may change, a branch decision may be made, and the instruction pointer is updated.

To match this causal structure when $x$ has control flow, we introduce latent branch decisions to our model family. The result is an RNN that processes some path through the program, with the path determined by these branch decisions. The simplest model is an oracle which has access to the ground-truth trace, a sequence $(n_0^*, n_1^*, \ldots)$ of the statement numbers generated by executing the program. Then, if the instruction pointer is chosen as $n_t = n_t^*$, the resulting model is an RNN over the ground truth trace of the program. We refer to this model as the Trace RNN.

If instead we model the branch decisions with a dense layer applied to the RNN's hidden state, then the instruction pointer is updated as

$$n_t = N_{\text{out}}(n_{t-1})|_j \qquad \text{where } j = \text{argmax} \, \text{Dense}(h_t). \tag{2}$$

This dense layer has two output units to predict which of the two branches is taken. We call this the Hard IP-RNN model (in contrast with the soft instruction pointer based models in Section 4.2). It is a natural model that respects the causal structure of a classical interpreter, but is not differentiable. A fully differentiable continuous relaxation will serve as our main model, the IPA-GNN. The information flows of each of these models, as well as that of a gated graph neural network, are shown in Figure 2.

### 4.2 Instruction Pointer Attention Graph Neural Network

To allow for fully differentiable end-to-end training, we introduce the Instruction Pointer Attention Graph Neural Network (IPA-GNN) as a continuous relaxation of the Hard IP-RNN. Rather than discrete branch decisions, the IPA-GNN makes soft branch decisions; a soft branch decision is a distribution over the possible branches from a particular statement. To accommodate soft branch decisions, we replace the instruction pointer of the Instruction Pointer RNN models ($n_t$) with a soft instruction pointer $p_{t,n}$, which at step $t$ is a distribution over all statements $x_n$. For each statement $x_n$ in the support of $p_{t,:}$, we might want the model to have a different representation of the program state. So, we use a different hidden state for each time step and statement. That is, the hidden state $h_{t,n} \in \mathbb{R}^H$ represents the state of the program, assuming that it is executing statement $n$ at time $t$.

As with the IP-RNN models, the IPA-GNN emulates executing a statement with an RNN over the statement's representation. This produces a state proposal $a_{t,n}^{(1)}$ for each possible current statement $n$

$$a_{t,n}^{(1)} = \text{RNN}(h_{t-1,n}, \text{Embed}(x_n)). \tag{3}$$

When executing straight-line code ($|N_{\text{out}}(x_n)| = 1|, n \to n'$) we could simply have $h_{t,n'} = a_{t,n}^{(1)}$, but in general we cannot directly use the state proposals as the new hidden state for the next statement, because there are sometimes multiple possible next statements ($|N_{\text{out}}(x_n)| = 2|$). Instead, the model computes a soft branch decision over the possible next statements and divides the state proposal $a_{t,n}^{(1)}$ among the hidden states of the next statements according to this decision. When $|N_{\text{out}}(x_n)| = 1$, the soft branch decision $b_{t,n,n'} = 1$, and when $|N_{\text{out}}(x_n)| = 2$, write $N_{\text{out}}(x_n) = \{n_1, n_2\}$ and

$$b_{t,n,n_1}, b_{t,n,n_2} = \text{softmax}\left(\text{Dense}(a_{t,n}^{(1)})\right). \tag{4}$$

All other values of $b_{t,n,:}$ are 0. The soft branch decisions determine how much of the state proposals $a_{t,n}^{(1)}$ flow to each next statement according to

$$h_{t,n} = \sum_{n' \in N_{\text{in}}(n)} p_{t-1,n'} \cdot b_{t,n',n} \cdot a_{t,n}^{(1)}. \tag{5}$$

A statement contributes its state proposal to its successors in an amount proportional both to the probability assigned to itself by the soft instruction pointer, and to the probability given to its successor by the branch decision. The soft branch decisions also control how much probability mass flows to each next statement in the soft instruction pointer according to

$$p_{t,n} = \sum_{n' \in N_{\text{in}}(n)} p_{t-1,n'} \cdot b_{t,n',n}. \tag{6}$$

The execution, branch, and aggregation steps of the procedure are illustrated in Figure 3.

By following the principle of respecting the causal structure of a classical interpreter, we have designed the IPA-GNN model. We hypothesize that by leveraging this causal structure, our model will exhibit better systematic generalization performance than models not respecting this structure.

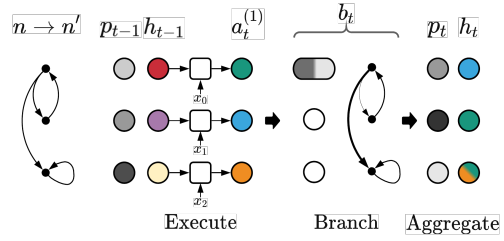

Figure 3: **A single IPA-GNN layer.** At each step of execution of the IPA-GNN, an RNN over the embedded source at each line of code produces state proposals $a_{t,n}^{(1)}$. Distinct state values are shown in distinct colors. A two output unit dense layer produces branch decisions $b_{t,n,:}$, shown as pill lightness and edge width. These are used to aggregate the soft instruction pointer and state proposals to produce the new soft instruction pointer and hidden states $p_{t,n}$ and $h_{t,n}$.

### 4.3 Relationship with IP-RNNs

As a continuous relaxation of the Hard IP-RNN model, the IPA-GNN bears a close relationship with the Instruction Pointer RNN models. Under certain conditions, the IPA-GNN is equivalent to the Hard IP-RNN model, and under still further conditions it is equivalent to the Trace RNN and Line-by-Line RNN models. Specifically:

- If the IPA-GNN's soft branch decisions saturate, the IPA-GNN and Hard IP-RNN are equivalent.
- If the Hard IP-RNN makes correct branch decisions, then it is equivalent to the Trace RNN.
- If the program $x$ is straight-line code, then the Trace RNN is equivalent to the Line-by-Line RNN.

To show the first two we can express the Hard IP-RNN and Trace RNN models in terms of two dimensional state $h_{t,n}$ and soft instruction pointer $p_{t,n}$. As before, let $N_{\text{out}}(x_n) = \{n_1, n_2\}$. Using this notation, the Hard IP-RNN's branch decisions are given by

$$b_{t,n,n_1}, b_{t,n,n_2} = \text{hardmax}\left(\text{Dense}(a_{t,n}^{(1)})\right), \tag{7}$$

where $\text{hardmax}$ projects a vector $v$ to a one-hot vector, i.e., $(\text{hardmax}\, v)|_j = 1$ if $j = \text{argmax}\, v$ and 0 otherwise. The Trace RNN's branch decisions in this notation are given by

$$b_{t,n,n'} = \mathbb{1}\{n_{t-1}^* = n \,\wedge\, n_t^* = n'\}. \tag{8}$$

Table 1: The IPA-GNN model is a message passing GNN. Selectively replacing its components with those of the GGNN yields two baseline models, NoControl and NoExecute. Blue expressions originate with the IPA-GNN, and orange expressions with the GGNN.

| | IPA-GNN (Ours) | NoControl | NoExecute | GGNN |
|---|---|---|---|---|
| $h_{0,n}$ | $= 0$ | $= 0$ | $= \text{Embed}(x_n)$ | $= \text{Embed}(x_n)$ |
| $a_{t,n}^{(1)}$ | $= \text{RNN}(h_{t-1,n}, \text{Embed}(x_n))$ | $= \text{RNN}(h_{t-1,n}, \text{Embed}(x_n))$ | $= h_{t-1,n}$ | $= h_{t-1,n}$ |
| $a_{t,n,n'}^{(2)}$ | $= p_{t-1,n'} \cdot b_{t,n',n} \cdot a_{t,n}^{(1)}$ | $= 1 \cdot a_{t,n}^{(1)}$ | $= p_{t-1,n'} \cdot b_{t,n',n} \cdot \text{Dense}(a_{t,n}^{(1)})$ | $= 1 \cdot \text{Dense}(a_{t,n}^{(1)})$ |
| $\tilde{h}_{t,n}$ | $= \sum\limits_{n' \in N_{\text{in}}(n)} a_{t,n,n'}^{(2)}$ | $= \sum\limits_{n' \in N_{\text{all}}(n)} a_{t,n,n'}^{(2)}$ | $= \sum\limits_{n' \in N_{\text{in}}(n)} a_{t,n,n'}^{(2)}$ | $= \sum\limits_{n' \in N_{\text{all}}(n)} a_{t,n,n'}^{(2)}$ |
| $h_{t,n}$ | $= \tilde{h}_t$ | $= \tilde{h}_t$ | $= \text{GRU}(h_{t-1,n}, \tilde{h}_{t,n})$ | $= \text{GRU}(h_{t-1,n}, \tilde{h}_{t,n})$ |

Otherwise the model definitions are the same as for the IPA-GNN. The final assertion follows from observing that straight-line code satisfies $n_t^* = t$.

These connections reinforce that the IPA-GNN is a natural relaxation of the IP-RNN models, and that it captures the causal structure of the interpreter it is based on. These connections also suggest there is a conceivable set of weights the model could learn that would exhibit strong generalization as a classical interpreter does if used in an unbounded execution setting.

## 4.4 Relationship with GNNs

Though the IPA-GNN is designed as a continuous relaxation of a natural recurrent model, it is in fact a member of the family of message passing GNN architectures. To illustrate this, we select the gated graph neural network (GGNN) architecture [18] as a representative architecture from this family. Table 1 illustrates the shared computational structure held by GGNN models and the IPA-GNN.

The GGNN model operates on the bidirectional form of the control flow graph. There are four possible edge types in this graph, indicating if an edge is a true or false branch and if it is a forward or reverse edge. The learned weights of the dense layer in the GGNN architecture vary by edge type.

The comparison highlights two changes that differentiate the IPA-GNN from the GGNN architecture. In the IPA-GNN, a per-node RNN over a statement is analogous to a classical interpreter's execution of a single line of code, while the soft instruction pointer attention and aggregation mechanism is analogous to the control flow in a classical interpreter. We can replace either the control flow components or the execution components of the IPA-GNN model with the expressions serving the equivalent function in the GGNN. If we replace both the control flow structures and the program execution components in the IPA-GNN, the result is precisely the GGNN model. Performing just the control flow changes introduces the NoControl model, while performing just the execution changes introduces the NoExecute model. We evaluate these intermediate models as baselines to better understand the source of the performance of the IPA-GNN relative to GNN models.

# 5 Experiments

Through a series of experiments on generated programs, we evaluate the IPA-GNN and baseline models for systematic generalization on the program execution as static analysis tasks.

**Dataset** We draw our dataset from a probabilistic grammar over programs using a subset of the Python programming language. The generated programs exhibit variable assignments, multi-digit arithmetic, while loops, and if-else statements. The variable names are restricted to v0, ..., v9, and the size of constants and scope of statements and conditions used in a program are limited. Complex nesting of control flow structures is permitted though. See the supplementary material for details.

From this grammar we sample 5M examples with complexity $c(x) \leq C$ to comprise $D_{\text{train}}$. For this filtering, we use program length as our complexity measure $c$, with complexity threshold $C = 10$. We then sample 4.5k additional samples with $c(x) > C$ to comprise $D_{\text{test}}$, filtering to achieve 500 samples each at complexities $\{20, 30, \ldots, 100\}$. An example program is shown in Figure 1.

For each complete program, we additionally construct partial programs by masking one expression statement, selected uniformly at random from the non-control flow statements. The target output remains the result of "correct" execution behavior of the original complete program.

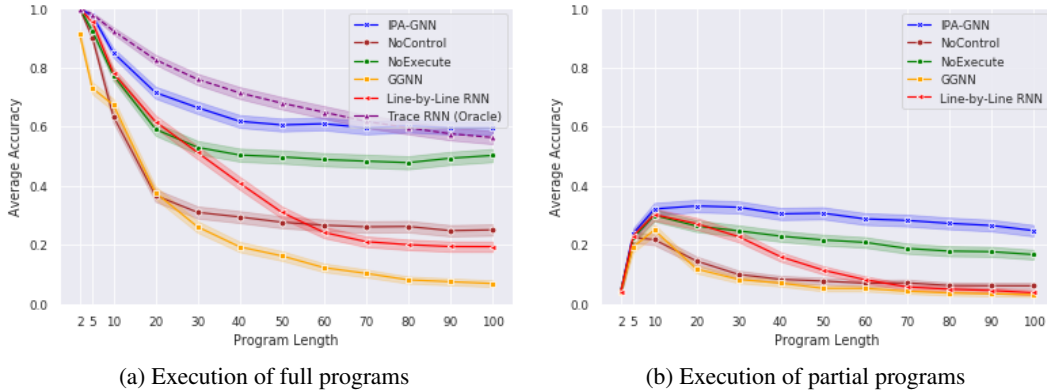

|                        |                         |
| :--------------------: | :---------------------: |
| (a) Execution of full programs | (b) Execution of partial programs |

Figure 4: Accuracy of models as a function of program length on the program execution tasks. Spread shows one standard error of accuracy.

In both the full program execution and partial program execution tasks, we let the target $y$ be the final value of v0 mod 1000. We select this target output to reduce the axes along which we are measuring generalization; we are interested in generalization to more complex program traces, which we elect to study independently of the complexity introduced by more complex data types and higher precision values. The orthogonal direction of generalization to new numerical values is studied in [22, 26]. We leave the study of both forms of generalization together to future work.

**Evaluation Criteria** On both tasks we evaluate the IPA-GNN against the Line-by-Line RNN baseline, R-GAT baseline, and the NoControl, NoExecute, and GGNN baseline models. On the full program execution task, we additionally compare against the Trace RNN model, noting that this model requires access to a trace oracle. Following Zaremba and Sutskever [29], we use a two-layer LSTM as the underlying RNN cell for the RNN and IPA-GNN models.

We evaluate these models for systematic generalization. We measure accuracy on the test split $D_{\text{test}}$ both overall and as a function of complexity (program length). Recall that every example in $D_{\text{test}}$ has greater program length than any example seen at training time.

**Training** We train the models for three epochs using the Adam optimizer [17] and a standard cross-entropy loss using a dense output layer and a batch size of 32. We perform a sweep, varying the hidden dimension $H \in \{200, 300\}$ and learning rate $l \in \{0.003, 0.001, 0.0003, 0.0001\}$ of the model and training procedure. For the R-GAT baseline, we apply additional hyperparameter tuning, yet we were unable to train an R-GAT model to competitive performance with the other models. For each model class, we select the best model parameters using accuracy on a withheld set of examples from the training split each with complexity precisely $C$.

**Program Execution Results** Table 2 shows the results of each model on the full and partial execution tasks. On both tasks, IPA-GNN outperforms all baselines.

Figure 4 breaks the results out by complexity. At low complexity values used during training, the Line-by-Line RNN model performs almost as well as the IPA-GNN. As complexity increases, however, the performance of all baseline models drops off faster than that of the IPA-GNN. Despite using the ground truth control flow, the Trace RNN does not perform as well as the IPA-GNN at all program lengths.

Table 2: Accuracies on $D_{\text{test}}$ (%)

| Model | Full | Partial |
| :--- | :--- | :--- |
| Trace RNN (Oracle) | 66.4 | — |
| Line-by-Line RNN | 32.0 | 11.5 |
| NoControl | 28.1 | 8.1 |
| NoExecute | 50.7 | 20.7 |
| GGNN | 16.0 | 5.7 |
| IPA-GNN (Ours) | **62.1** | **29.1** |

Examining the results of the ablation models, NoExecute significantly outperforms NoControl, indicating the importance of instruction pointer attention for the IPA-GNN model.

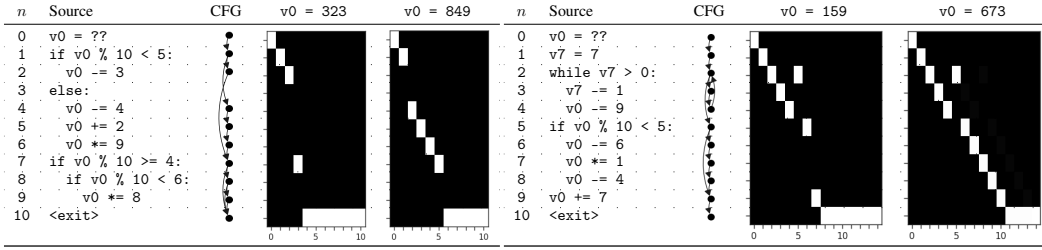

Figure 5: **Instruction Pointer Attention.** Intensity plots show the soft instruction pointer $p_{t,n}$ at each step of the IPA-GNN on two programs, each with two distinct initial values for v0.

Figure 5 shows the values of the soft instruction pointer $p_{t,n}$ over the course of four predictions. The IPA-GNN learns to frequently produce discrete branch decisions. The model also learns to short-circuit execution in order to produce the correct answer while taking fewer steps than the ground truth trace. From the first program, we observe the model has learned to attend only to the path through the program relevant to the program's result. From the second program, we see the model attends to the while-loop body only once, where the ground truth trace would visit the loop body seven times. Despite learning in a setting of bounded execution, the model learned a short-circuited notion of execution that exhibits greater systematic generalization than any baseline model.

## 6 Conclusion and Future Work

Following a principled approach, we designed the Instruction Pointer Attention Graph Neural Network architecture based on a classical interpreter. By closely following the causal structure of an interpreter, the IPA-GNN exhibits stronger systematic generalization than baseline models on tasks requiring reasoning about program execution behavior. The IPA-GNN outperformed all baseline models on both the full and partial program execution tasks.

These tasks, however, only capture a subset of the Python programming language. The programs in our experiments were limited in the number of variables considered, in the magnitude of the values used, and in the scope of statements permitted. Even at this modest level of difficulty, though, existing models struggled with the tasks, and thus there remains work to be done to solve harder versions of these tasks and to scale these results to real world problems. Fortunately the domain naturally admits scaling of difficulty and so provides a good playground for studying systematic generalization.

We believe our results suggest a promising direction for models that solve real world tasks like programming by example. We proffer that models like the IPA-GNN may prove useful for constructing embeddings of source code that capture information about a program's semantics.

## Broader Impact

Our work introduces a novel neural network architecture better suited for program understanding tasks related to program executions. Lessons learned from this architecture will contribute to improved machine learning for program understanding and generation. We hope the broader impact of these improvements will be improved tools for software developers for the analysis and authoring of new source code. Machine learning for static analysis produces results with uncertainty, however. There is risk that these techniques will be incorporated into tools in a way that conveys greater certainty than is appropriate, and could lead to either developer errors or mistrust of the tools.

## Acknowledgments and Disclosure of Funding

This work benefited from conversations and feedback from our colleagues. In particular we thank Daniel Johnson, Disha Shrivastava, Sarath Chandar, and Rishabh Singh for thoughtful discussions.

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
