[Supplementary Material]

## A Architecture Details

We provide additional architectural details here beyond those provided in the paper.

In this work, all GNN models (IPA-GNN, NoExecute, NoControl, GGNN, and R-GAT) compute their final hidden state as $h_{\text{final}} = h_{T(x),n_{\text{exit}}}$. Here $n_{\text{exit}}$ is the index of the program's exit statement, and the number of neural network layers $T(x)$ is computed as

$$T(x) = \sum_{0 \le i \le n_{\text{exit}}} 2^{\text{LoopNesting}(i)} + \sum_{i \in \text{Loops}(x)} 2^{\text{LoopNesting}(i)} \tag{9}$$

LoopNesting$(i)$ denotes the number of loops with loop-body including statement $x_i$. And Loops$(x)$ denotes the set of while-loop statements in $x$. This provides enough layers to permit message passing along each path through a program's loop structures twice, but not enough layers for the IPA-GNN to learn to follow the ground truth trace of most programs.

In all models, the output layer consists of the computation of logits, followed by a softmax cross-entropy categorical loss term. The softmax-logits are computed according to

$$s = \text{softmax}\left(\text{Dense}(h_{\text{final}})\right). \tag{10}$$

The cross entropy loss is then computed as

$$L = -\sum_{i}^{K} \mathbb{1}_{y=i} \log(s_i). \tag{11}$$

This loss is then optimized using a differentiable optimizer during training.

## B Data Generation

For the learning to execute full and partial programs tasks, we generate a dataset from a probabilistic grammar over programs. Figure 6 provides the grammar. `If`, `IfElse`, and `Repeat` statements are translated into their Python equivalents. `Repeat` statements are represented using a while-loop and counter variable selected from `v1`...`v9` uniformly at random, excluding those variables already in use at the entrance to the `Repeat` statement.

Attention plots for randomly sampled examples from the full program execution task are shown in Figure 7. We then mask a random statement in each example and run the partial program execution IPA-GNN model over each program, showing the resulting attention plots in Figure 8. All four tasks are solved correctly in the full program execution task task, and the first three are solved correctly in the partial execution task, while the fourth partial execution task shown is solved incorrectly.

$$
\begin{aligned}
\text{Program } P &:= I\ B \\
\text{Initialization } I &:= v_0\ \text{=}\ M \\
\text{Block } B &:= B\ S\ |\ S \\
\text{Statement } S &:= E\ |\ \texttt{If}(C, B)\ |\ \texttt{IfElse}(C, B_1, B_2)\ |\ \texttt{Repeat}(N, B) \\
&\quad\ |\ \texttt{Continue}\ |\ \texttt{Break}\ |\ \texttt{Pass} \\
\text{Condition } C &:= v_0\ \texttt{mod}\ 10\ O\ N \\
\text{Operation } O &:= \texttt{>}\ |\ \texttt{<}\ |\ \texttt{>=}\ |\ \texttt{<=} \\
\text{Expression } E &:= v_0\ \texttt{+=}\ N\ |\ v_0\ \texttt{-=}\ N\ |\ v_0\ \texttt{*=}\ N \\
\text{Integer } N &:= 0\ |\ 1\ |\ 2\ |\ \ldots\ |\ 9 \\
\text{Integer } M &:= 0\ |\ 1\ |\ 2\ |\ \ldots\ |\ 999
\end{aligned}
$$

Figure 6: Grammar describing the generated programs comprising the dataset in this paper.

**Figure 7 (top-left panel)**

| n | Source | CFG | IPA |
|---|--------|-----|-----|
| 0 | v0 = 589 | | |
| 1 | if v0 % 10 >= 8: | | |
| 2 |   v0 *= 4 | | |
| 3 | else: | | |
| 4 |   if v0 % 10 < 0: | | |
| 5 |     v0 *= 1 | | |
| 6 |   else: | | |
| 7 |     if v0 % 10 >= 6: | | |
| 8 |       if v0 % 10 < 3: | | |
| 9 |         v0 += 9 | | |
| 10 | <exit> | | |

**Figure 7 (top-right panel)**

| n | Source | CFG | IPA |
|---|--------|-----|-----|
| 0 | v0 = 36 | | |
| 1 | if v0 % 10 >= 7: | | |
| 2 |   v0 *= 3 | | |
| 3 |   if v0 % 10 > 3: | | |
| 4 |     v0 *= 4 | | |
| 5 |     v5 = 3 | | |
| 6 |     while v5 > 0: | | |
| 7 |       v5 -= 1 | | |
| 8 |       break | | |
| 9 | v0 *= 2 | | |
| 10 | <exit> | | |

**Figure 7 (bottom-left panel)**

| n | Source | CFG | IPA |
|---|--------|-----|-----|
| 0 | v0 = 528 | | |
| 1 | v0 *= 1 | | |
| 2 | v0 += 9 | | |
| 3 | v0 += 3 | | |
| 4 | if v0 % 10 < 8: | | |
| 5 |   if v0 % 10 < 3: | | |
| 6 |     if v0 % 10 < 0: | | |
| 7 |       v0 -= 7 | | |
| 8 |       v0 -= 9 | | |
| 9 | <exit> | | |

**Figure 7 (bottom-right panel)**

| n | Source | CFG | IPA |
|---|--------|-----|-----|
| 0 | v0 = 117 | | |
| 1 | if v0 % 10 <= 6: | | |
| 2 |   v0 -= 9 | | |
| 3 |   v0 += 7 | | |
| 4 | else: | | |
| 5 |   v1 = 2 | | |
| 6 |   while v1 > 0: | | |
| 7 |     v1 -= 1 | | |
| 8 |     v0 -= 6 | | |
| 9 | v0 *= 1 | | |
| 10 | <exit> | | |

Figure 7: Intensity plots show the soft instruction pointer $p_{t,n}$ at each step of the IPA-GNN during full program execution for four randomly sampled programs.

**Figure 8 (top-left panel)**

| n | Source | CFG | IPA |
|---|--------|-----|-----|
| 0 | v0 = 589 | | |
| 1 | if v0 % 10 >= 8: | | |
| 2 |   v0 *= 4 | | |
| 3 | else: | | |
| 4 |   if v0 % 10 < 0: | | |
| 5 |     v0 *= 1 | | |
| 6 |   else: | | |
| 7 |     if v0 % 10 >= 6: | | |
| 8 |       if v0 % 10 < 3: | | |
| 9 |         [MASK] | | |
| 10 | <exit> | | |

**Figure 8 (top-right panel)**

| n | Source | CFG | IPA |
|---|--------|-----|-----|
| 0 | v0 = 36 | | |
| 1 | if v0 % 10 >= 7: | | |
| 2 |   [MASK] | | |
| 3 |   if v0 % 10 > 3: | | |
| 4 |     v0 *= 4 | | |
| 5 |     v5 = 3 | | |
| 6 |     while v5 > 0: | | |
| 7 |       v5 -= 1 | | |
| 8 |       break | | |
| 9 | v0 *= 2 | | |
| 10 | <exit> | | |

**Figure 8 (bottom-left panel)**

| n | Source | CFG | IPA |
|---|--------|-----|-----|
| 0 | v0 = 528 | | |
| 1 | [MASK] | | |
| 2 | v0 += 9 | | |
| 3 | v0 += 3 | | |
| 4 | if v0 % 10 < 8: | | |
| 5 |   if v0 % 10 < 3: | | |
| 6 |     if v0 % 10 < 0: | | |
| 7 |       v0 -= 7 | | |
| 8 |       v0 -= 9 | | |
| 9 | <exit> | | |

**Figure 8 (bottom-right panel)**

| n | Source | CFG | IPA |
|---|--------|-----|-----|
| 0 | v0 = 117 | | |
| 1 | if v0 % 10 <= 6: | | |
| 2 |   v0 -= 9 | | |
| 3 |   v0 += 7 | | |
| 4 | else: | | |
| 5 |   v1 = 2 | | |
| 6 |   while v1 > 0: | | |
| 7 |     v1 -= 1 | | |
| 8 |     [MASK] | | |
| 9 | v0 *= 1 | | |
| 10 | <exit> | | |

Figure 8: The same programs as in Figure 7, with a single statement masked in each. The intensity plots show the soft instruction pointer $p_{t,n}$ at each step of the IPA-GNN during partial program execution.