[Reviews · NeurIPS 2020]

Review 1

Summary and Contributions: This paper proposes a new NN architecture for the task of learning to execute programs. The proposed architecture (IPA-GNN) uses recurrent NNs as well as message passing in GNNs to model attending to a given instruction at time t to learn how a program with control flow is executed. It handles control flow by means of a soft attention mechanism and execution by means of a soft instruction pointer. Their results show that they can achieve better accuracies as well as generalization to larger programs compared to existing techniques in the literature.

Strengths: * The authors develop their model similar to a basic interpreter. Their model is a natural description of what a soft interpreter would look like. * They train on programs with lower complexity (lines of code) and test on programs with larger complexity, especially with the aim of showing generalizability.

Weaknesses: ** Training methodology is not properly mentioned. * What loss function did you use? It is a regression-based loss on the numerical value? or a cross-entropy loss on the 1000 possible values for v_0 mod 1000? * What is the accuracy you report in Table 2? Is it the % of the time the result 100% matched the target? or is it a regression loss (something like MAPE)? ** Need more comparison to other GNNs * one explicitly relevant GNN architecture is the "Graph Attention Network". It would be good if you can mention how it is related to IPA-GNN. GATs also allow attending using different weights to incoming messages from neighbors. Note that GAT is a kind of convolution-based GNN and does not use a recurrent unit, so you will have to adapt the attention mechanism in the context of GGNN. * GraphSAGE allows different aggregation functions. In IPA-GNN you use (+) as your aggregator. Did you try out different aggregators? ** Claim that GGNN = NoExecute and NoControl model * GGNN is performing Non-linear(linear(hidden_states)) where as IPA-GNN is performing Linear(Non-linear(hidden_states)). Essentially, GGNN is applying the non-linearity after the messages are aggregated, but in your case, the non-linearity is applied first and then the messages are passed and aggregated. It is not clear to me, that these two computational patterns are equivalent. It would be good if you can make a formal argument on this. ** Program generator * How realistic are the programs generated by your probabilistic grammar? ********** Post rebuttal comments ************************ I would still like a more formal argument about the equality between GGNN and no execute / no control model, since it is still not clear how linear(non-linear(hidden_states)) == non-linear(linear(hidden_states)). However, I am positive about the rest of the paper and am keeping my original score.

Correctness: * I am not sure about the claim that NoExecute + NoControl IPA-GNN is equivalent to GGNN * The calculations of accuracies in Table 2 and Figure 2 are not properly mentioned.

Clarity: * The paper is well-written. The symbols and math are easy to understand.

Relation to Prior Work: * I am happy with the related work mentioned in the context of learning to execute. I would like to see more comparisons with other GNN architectures proposed in literature.

Reproducibility: Yes

Additional Feedback: * Please address my concerns in the weaknesses section. * How susceptible is your system to syntactic changes? I am curious to know if a pure renaming of the variables would result in the same output.


Review 2

Summary and Contributions: The paper intends to achieve the best of both worlds -- GNN and RNN -- on exploring program structure and reasoning long program sequence for static analysis. To this end, the paper proposes a GNN architecture, called Instruction Pointer Attention Graph Neural Network (IPA-GNN). The key idea is to develop an instruction attention GNN model with RNN as the interpreter over statement representation of latent branch decisions. Testing of learning to execute full programs with control flow graphs and partial programs shows that the proposed model outperforms both RNN and GNN only models.

Strengths: The study identifies that an RNN trace model with latent branch decisions is a special case of a GNN. The proposed model outperforms both RNN and GNN models on the task of learning to execute programs.

Weaknesses: It would be good to have more evaluations on each component of IPA-GNN.

Correctness: Yes.

Clarity: Certain part of the paper is hard to read, e.g., the descriptions of instruction pointer RNN and IPA-GNN make it confusing to interpret the relationship between the two.

Relation to Prior Work: Yes.

Reproducibility: Yes

Additional Feedback: The paper tackles a real problem in program analysis especially branches and addresses the tradeoff between GNN and RNN. It appears promising that the proposed IPA-GNN is demonstrated to be more efficient than both GNN and RNN by identifying an effective way to combine the benefits of the two. The proposed model is evaluated by a large dataset with various branch structures. But it would be good to also evaluate the effect of each model component in addition to the overall performance. The description of the model can also be improved by showing an overview diagram or graph. Comments after rebuttal: Most of the reviewer's concerns were addressed by the rebuttal.


Review 3

Summary and Contributions: The authors propose a novel GNN architecture, the Instruction Pointer Attention Graph Neural Network (IPA-GNN) to handle the task of learning to execute programs using control flow graphs. Results show that the IPA-GNN outperforms a variety of RNN and GNN baselines on both the full and partial program execution tasks.

Strengths: The authors introduced two variants of the "learning to execute" task: full and partial program execution.

Weaknesses: The authors developed the Instruction Pointer Attention Graph Neural Network (IPA-GNN). While the work is named as an attention model, the authors barely talked about the attention mechanism directly. The authors choose to compare GGNN. GGNN is a general model not specifically designed for these tasks. And the GGNN shows very bad performance in Fig.2. What is the reason to select this model? The authors did not evaluate many other methods mentioned in related work.

Correctness: yes

Clarity: ok

Relation to Prior Work: yes

Reproducibility: Yes

Additional Feedback: The model architecture improvement in this paper is not strong enough. ============== POST REBUTTAL The new baseline model is a worthy addition to the experimental part. The author mentioned a significant improvement compared to line-by-line RNN. The performance achieved by three epochs training (line 286 training section). Limited the training epochs to 3 is strange, especially when the training set has 5M examples. Overall, based on the improvement, I change my score to 5.

[Author Response · NeurIPS 2020]

Thank you for the thorough reviews and insightful comments.

**Clarifications** We first address some of the smaller points raised by the reviewers. R1, we clarify that the loss function (line 287) is indeed a cross-entropy loss on the 1000 possible values for $v_0 \% 1000$. R1, the accuracy in Table 2 is (just as you surmised) the percent of examples in the test set that the model produces the exact correct result for. Recall that the test set has programs with lengths 20 through 100, which places them outside of the train distribution. R3, we clarify that the "soft instruction pointer" is the novel attention mechanism that the paper draws its title from. We discuss it in detail in Sections 4.2-4.4 and Section 5. We have revised the paper to clarify each of these points. R2, we like your suggestion of adding an overview diagram and will gladly make this addition for the camera-ready if accepted.

**Why GGNN? And a new baseline: R-GAT** R3, the primary reason we elect in the paper to compare against the GGNN model rather than e.g. GAT or GraphSage is that GGNN is a more common model choice for program understanding tasks [1, 4–8]. Additionally Table 3 of [2] compares variants of common message passing GNN architectures (GGNN, GCN, and GAT) on the VariableMisuse task and finds the degree of variation in performance to be considerably smaller than that between the GGNN and IPA-GNN on learning to execute. So, we do not think that the comparatively small differences in these GNN architectures will yield significant differences on the learning to execute tasks.

Nevertheless, we take R1 and R3's suggestion to introduce a new baseline, R-GAT [3] (non-relational GAT lacks edge-types and so cannot distinguish between true and false branches of if statements). Preliminary results suggest similar performance to the GGNN. We will incorporate the R-GAT baseline results into our paper if accepted.

**"NoControl" model equation** R1 asked about our claim that "GGNN = NoExecute and NoControl". Indeed, we made a typographic error in Table 1, and we thank R1 for catching our mistake. The correct equation for $a_{t,n,n'}^{(2)}$ for the NoControl model is simply $a_{t,n,n'}^{(2)} = a_{t,n}^{(1)}$. Please note the difference between this and what is printed in Table 1 in the paper. We checked and the error is purely typographic, not an error with the experiments themselves. We have now corrected this, and we hope this resolves your concern. With this fix in place, it should be apparent that the NoControl and NoExecute models interpolate between the IPA-GNN and GGNN model. When both the NoControl and NoExecute changes are simultaneously applied to the IPA-GNN model definition, the result is the GGNN model. R2, we hope this correction also addresses your request for improved evaluation of the individual components of the IPA-GNN. We stress for R2 that the primary purpose of the NoControl and NoExecute models is to determine the significance of the individual components of the IPA-GNN model. These models form an ablation study of sorts, but rather than *ablating* the individual components of the IPA-GNN model (which would render it useless) we replace them with the analogous component from the GGNN model. So we think of this more as an "interpolation" study than an "ablation" study, but it serves the same purpose.

**Significance of improvement** We note a disagreement between the reviewers about the significance of the improvement our architecture contributes ($26.7\% \to 43.8\%$ and $17.2\% \to 28.3\%$ out-of-distribution accuracy on full and partial program execution respectively). R3 notes that this improvement is not strong enough. We think this improvement is more than enough to warrant publication, and respectfully request that R3 reconsider their stance. We hope the reviewers get to discuss this point in their post author response discussions.

# References

[1] Miltiadis Allamanis, Marc Brockschmidt, and Mahmoud Khademi. Learning to represent programs with graphs, 2017.

[2] Marc Brockschmidt. {GNN}-fi{lm}: Graph neural networks with feature-wise linear modulation, 2020. URL https://openreview.net/forum?id=HJe4Cp4KwH.

[3] Dan Busbridge, Dane Sherburn, Pietro Cavallo, and Nils Y. Hammerla. Relational graph attention networks, 2019. URL https://openreview.net/forum?id=Bklzkh0qFm.

[4] Mingzhe Li, Jianrui Pei, Jin He, Kevin Song, Frank Che, Yongfeng Huang, and Chitai Wang. Using ggnn to recommend log statement level, 2019.

[5] Yujia Li, Daniel Tarlow, Marc Brockschmidt, and Richard Zemel. Gated graph sequence neural networks, 2015.

[6] Mingming Lu, Dingwu Tan, Naixue Xiong, Zailiang Chen, and Haifeng Li. Program classification using gated graph attention neural network for online programming service, 2019.

[7] Jessica Schrouff, Kai Wohlfahrt, Bruno Marnette, and Liam Atkinson. Inferring javascript types using graph neural networks, 2019.

[8] Sahil Suneja, Yunhui Zheng, Yufan Zhuang, Jim Laredo, and Alessandro Morari. Learning to map source code to software vulnerability using code-as-a-graph, 2020.


[Meta-Review · NeurIPS 2020]

All reviewers agree there is novelty here, and the domain of application (program analysis and understanding) is of growing importance at NeurIPS. The additional baseline comparisons mentioned in the rebuttal are welcome.